# Temporal and Spatial Distribution of the Toxic Epiphytic Dinoflagellate *Ostreopsis* cf. *ovata* in the Coastal Waters off Jeju Island, Korea

**Jaeyeon Park \*, Jinik Hwang, Jun-Ho Hyung and Eun Young Yoon**

Advanced Institute of Convergence Technology, Suwon 16229, Korea; jinike12@snu.ac.kr (J.H.);
hjh1120@snu.ac.kr (J.-H.H.); journal04@snu.ac.kr (E.Y.Y.)
**\*** Correspondence: bada0@snu.ac.kr; Tel.: +82-31-888-9042; Fax: +82-31-888-9040

**Abstract:** The temporal and spatial distribution of the toxic epiphytic dinoflagellate *Ostreopsis* cf. *ovata* was investigated off the Jeju coastal waters, Korea, from July 2016 to January 2019. The results showed that the presence of *Ostreopsis* cf. *ovata* in 184 macroalgae was 79.3%, and it was more frequently attached to red algae and brown algae than to green algae. The abundance of *Ostreopsis* cf. *ovata* as determined by quantitative real-time polymerase chain reactions (qPCR) and microscopic analysis was 4–3204 cells $g^{-1}$, and the maximum abundance observed in September 2018, when the water temperature was 24.4 °C. The abundance was higher in summer and autumn than in spring and winter. Spatially, high abundance was observed in autumn on the northern coast of Jeju Island and, in summer, in the southern and eastern coastal waters. The water temperature of Jeju coastal waters in winter remained higher than 15 °C, and this species could be overwintering in the Jeju waters. Therefore, further monitoring and research are needed to evaluate the proliferation of *Ostreopsis* cf. *ovata*, which contains a novel toxin with unidentified effects on humans.

**Keywords:** epiphytic dinoflagellate; *ostreopsis* cf. *ovata*; abundance; temporal and spatial distribution; Jeju island

## 1. Introduction

The genus *Ostreopsis* is one of the toxic epiphytic marine dinoflagellates whose geographical distribution has been expanding from tropical to temperate region [1–3]. The genus *Ostreopsis* produces the palytoxin (PLTX) complex which is one of the most potent biotoxins [4–6] and causes human food poisoning by bioconcentration in seafood. Four *Ostreopsis* species—*O.* cf. *ovata*, *O. lenticularis*, *O. mascarenensis*, and *O. siamensis*—produce PLTX-like compounds. Blooms of PLTX-producing algae cause serious problems in aquatic ecosystem and fisheries and are dangerous to human health [7,8]. Recently, global warming has been shown to affect the marine ecosystem structure and promote outbreaks of harmful algal blooms with the consequence production of PLTX caused by *Ostreopsis* species [9]. Especially, *Ostreopsis* cf. *ovata* produces a wide range of PLTX-like compounds and ovatoxins (ovatoxin, a~f) [10–12], among which ovatoxin is a non-PLTX compound recently reported as a marine toxin affecting shrimp [13].

In previous decades, *Ostreopsis* blooms were common in temperate water, mostly in the Mediterranean Sea during summer to autumn and other temperate areas (Table 1). It has also been reported that this genus has high tolerance of a wide range of temperature. [14].

Blooms of *Ostreopsis* species cause serious human health problems via inhalation of sea water droplets containing *Ostreopsis* cells or aerosolized toxins [6,15,16]. *Ostreopsis* can create and proliferate mucus floating clusters. These clusters float on the water surface, and toxins released from the cells

through cell lysis can be aerosolized sea spray [6,17–19]. In addition, these organisms play an important role in the marine food chain and are a feed source for large protozoa or shellfish through filter feeding. Consumption of epiphytic dinoflagellates including *Ostreopsis* induces paralytic shellfish poisoning, diarrhea shellfish poisoning, and neurotoxic shellfish poisoning [11]. Thus, the dinoflagellate toxin and its analogs can be transferred to humans. Respiratory symptoms from sea spray were first reported in 2003 Italy. The abundance of *Ostreopsis* was more than $2.5 \times 10^6$ cells $\mathrm{g}^{-1}$ fw of macroalgae, and high cell concentration was detected in the water column. Two hundred people exposed to marine aerosols received local first aid treatment, and among them, 20 people required extended hospitalization [16]. Furthermore, the distribution of *Ostreopsis* species from 1985 to 2017 has been reported to occur in more two million cells $\mathrm{g}^{-1}$ in the Mediterranean at temperature ranged >20 °C and salinity ranged >30 (Table 1).

In addition, field studies reported that environmental factors such as salinity and temperature can play important role in promoting the spread of *Ostreopsis* spp. [20,21]. However, some studies report that proliferation can differ depending on the environmental conditions and on species and geographical requirements [22–25]. As a result, an increasing number of studies have been performed to evaluate the role of each environmental variable and geographic impact on growth and toxicity.

Jeju Island is located in temperate region, but its coastal waters are affected by the Tsushima Warm Current, which introduces tropical fish and invertebrates [26]. Moreover, epiphytic dinoflagellates that have appeared in tropical to subtropical region have been found in Jeju coastal waters since 2010. Potentially toxic dinoflagellates including *Amphidinium carterae*, *Coolia monotis*, *Gambierdiscus toxicus*, and *Ostreopsis* sp. were found in Jeju coastal waters [27,28]. However, the temporal and spatial distribution of *Ostreopsis* cf. *ovata* off Jeju coastal waters is not well-understood.

For understanding the presence status of *O.* cf. *ovata* in Jeju coastal waters, we investigated the spatial and seasonal distributions of the *O.* cf. *ovata* in Jeju, Korea. The abundance of this species was quantified in macroalgae samples collected from four sites at Jeju Island in four seasons from July 2016 to January 2019 by quantitative real-time polymerase chain reactions (qPCR) and direct microscopic-counting methods. This study also provides a basis for understanding effects of critical environmental factor such as temperature and salinity on the spatial and temporal distributions of *O.* cf. *ovata*.

Potentially toxic benthic algae have been widely evaluated in recent decades. There are concerns regarding the damage caused by benthic dinoflagellates. Thus, the aim of this research was to prevent future damage by toxic epiphytic dinoflagellates by monitoring their appearance and patterns of distribution in Korean waters.

**Table 1.** The abundance of *Ostreopsis* species in various locations obtained from the literature.

| Date | Species | Origin | Abundance (Cells L⁻¹) | Abundance (Cells g⁻¹) | Temperature (°C) | Salinity | Reference |
|---|---|---|---|---|---|---|---|
| 1985 October | *Ostreopsis lenticularis* | Caribbean Sea, Puerto Rico | | 16,000 | >28 | | [29] |
| 1995 | *Ostreopsis* sp. | Mediterranean, Catalan Sea | 78,000 | | | | [30] |
| 1997 July | *Ostreopsis* sp. | Mediterranean, Catalan Sea | | 596,000 | 24–26 | 37.2–38.1 | [31] |
| 1998 August | *Ostreopsis* cf. *ovata* | Mediterranean, Ligurian Sea | 50,000 | | 25 | | [32] |
| 2001 July | *Ostreopsis ovata* | Islands of Hawaii | | 7346 | 25.4 | 26.7 | [33] |
| 2001 August–September | *Ostreopsis siamensis* | Gulf of Tunis, Tunisia | | 3600 | 20–27 | 37 | [34] |
| 2001 November | *Ostreopsis* sp. | Islands of Hawaii | | 18,194 | 27.6 | 30.8 | [33] |
| 2001–2002 | *Ostreopsis lenticularis* | northernwestern Cuba | | <1000 | | | [35] |
| 2002 August | *Ostreopsis* cf. *ovata* | Mediterranean, Tyrrhenian Sea | 10,550 | | | | [36] |
| 2004 February | *Ostreopsis siamensis* | New Zealand | | 1,406,000 (±385,500) | 20 | | [37] |
| 2005 May–December | *Ostreopsis heptagona* | Gulf of Mexico | | 1202 | 29.5 | 31 | [38] |
| 2006 July | *Ostreopsis ovata* | Mediterranean, Ligurian Sea | 87,000 (±27,000) | 2,541,000 (±588,000) | >26 | 38.0–38.2 | [16] |
| 2007 October | *Ostreopsis* cf. *ovata* | Mediterranean, Adriatic Sea | 25,200 | 1,700,000 | 16.8–21.8 | | [22] |
| 2008 June–August | *Ostreopsis* cf. *ovata* | Mediterranean, Balearic Sea | 1000 | 60,000 | 23–27.5 | | [21] |
| 2008 July | *Ostreopsis* cf. *ovata* | Mediterranean, Ligurian Sea | 213,000 | 600,000 | 22.5 | | [20] |
| 2008 September | *Ostreopsis* spp. | Northwestern Sea of Japan | | 4213 | 17–22.2 | | [39] |
| 2009 October | *Ostreopsis* spp. | Jeju Island, Korea | | 8660 | 21.0–23.6 | 28.9–32.5 | [28] |
| 2010 July–August | *Ostreopsis* spp. | Mediterranean, Tyrrhenian Sea | 21,680 | 79,000 | 19.8–25.8 | 36.5–37.9 | [40] |
| 2010 September–October | *Ostreopsis* cf. *ovata* | Mediterranean, Adriatic Sea | 42,600 | 334,306 | | | [41] |
| 2013 September | *Ostreopsis* spp. | Lagos Bay, Atlantic Ocean | 17,000 | 115,831 | 24 | | [42] |
| 2015 July | *Ostreopsis* cf. *ovata* | Mediterranean, Ligurian Sea | 51,719 | 2,289,100 | 25.9 | | [43] |
| 2016 October | *Ostreopsis* spp. | Lagos Bay, Atlantic Ocean | 640 | 45,251 | 20 | | [42] |
| 2017 July | *Ostreopsis* sp. | Korea (Pohang) | | 1588 | 20–25 | 29.1–32.1 | [44] |
| 2017 June–July | *Ostreopsis* cf. *ovata* | Mediterranean, Ligurian Sea | 81,380 | 890,528 | >25 | | [45] |

## 2. Materials and Methods

### 2.1. Sample Collection and Treatment

Macroalgae samples were collected by scuba divers from July 2016 to January 2019 within 10 m of water depth. Four different macroalgae species were collected at the four sampling sites (Jeju—north, Gosan—west, Wimi—south, Seongsan—east) off Jeju Island (Figure 1). Environmental factors, such as water temperature and salinity were measured in the surface water using a YSI Professional plus (Xylem, Inc., Rye Brook, USA). The collected macroalgae were morphologically identified using an illustrated book [46]. Epiphytic dinoflagellates attached to the macroalgae were collected as described by Baek [47]. The collected macroalgae were transferred into a 1-L bottle, which as filled with filtered seawater, and then shaken vigorously approximately 200 times to detach the epiphytic dinoflagellates. After filtering the sample through a 100-μm mesh to separate the macroalgae from seawater, the macroalgae were rapidly frozen on dry ice and then transported to the laboratory. A 300-mL sample containing epiphytic dinoflagellates was fixed with formalin (final concentration 1%), and another 300-mL sample was transferred to the laboratory as a live sample for establishing strains of epiphytic dinoflagellates.

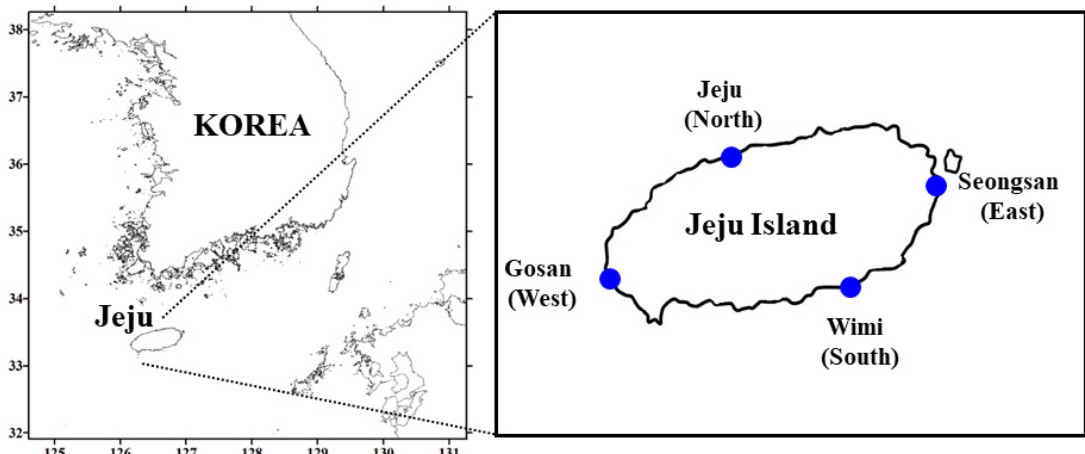

**Figure 1.** Maps of sampling stations (Jeju (north), Gosan (west), Wimi (south), and Seongsan (east) in Jeju Island, Korea).

To obtain a DNA sample, 20–50 mL seawater sample obtained by shaking the macroalgae was filtered through a GF/C filter and the filter paper was rapidly frozen on dry ice. Macroalgae were transported to the laboratory and weighed after identification.

### 2.2. Fluorescence Microscopy

The fixed samples were concentrated 5–6 times for microscopic observation. The fixed 300 mL of sample was left still overnight to allow the cells to sink, and the supernatant was removed to adjust the final volume to 50–60 mL. In addition, the frozen macroalgae were weighed fresh after remove excess seawater. A solution of Calcofluor (Sigma-Aldrich, St. Louis, MO, USA) was added to the fixed sample at a final concentration of 10 μg/mL and stained for 1–2 min in the dark. The cells were counted in a Sedgwick–Rafter chamber (SPI Supplies, West Chester, PA, USA) at 100X magnification using an epi-fluorescence microscope (BX 53, Olympus, Tokyo, Japan). The cells were photographed with a DP73 digital camera system (Olympus, Tokyo, Japan).

The final concentration of epiphytic dinoflagellates attached to each macroalgae was calculated by calculating the adhesion density per gram of fresh weight of macroalgae (cells $g^{-1}$), as described by Ishikawa and Kurashima [48].

### 2.3. Culture, DNA Extraction, and PCR Amplification

To establish a single cell strain of *O.* cf. *ovata*, the live sample transferred to the laboratory was placed in a six-well plate, and single cell isolation was performed under a dissecting microscope (SZX10, Olympus, Tokyo, Japan). Clonal cultures of *O.* cf. *ovata* were established by two serial single-cell isolations. After sufficient growth, the cells were transferred into a 30-mL flask. Once dense cultures of *O.* cf. *ovata* were obtained, the cells were transferred into 200-mL PC bottles containing fresh f/2 seawater media. The DNA sequences of these cells were analyzed when the concentration of *O.* cf. *ovata* was more than $10^5$ cells ml$^{-1}$. The dense culture (1 mL) was centrifuged and the pellet was used for DNA extraction. DNA extraction was carried out with an AccuPrep Genomic DNA Extraction Kit (BIONEER, Daejeon, Korea), and then the SSU-ITS-LSU section of rDNA was amplified using universal eukaryotic primers [49]. The obtained rDNA sequence were confirmed by BLAST (https://blast.ncbi.nlm.nih.gov/ Blast.cgi, NCBI).

To obtain DNA from field samples, 20–50 mL of samples obtained by shaking the macroalgae which collected from the sampling sites in the Jeju coastal waters were filtered with Whatman GF/C (pore size 1.2 μm), and genomic DNA (gDNA) was extracted with an AccuPrep Genomic DNA Extraction Kit according to the manufacturer's protocol.

### 2.4. Design of Species-Specific Primers

After obtaining the total sequences of rDNA from the *O.* cf. *ovata* strain grown under laboratory conditions, *O.* cf. *ovata*-specific primers (forward: 5′-GGCCATTCCTAAGGACATCA−3′, reverse: 5′-TGGCCATATACAGCATGTTGAC−3′), and a TaqMan probe (5′-ATCATGCATTGTGTGAGTGTGTG ATGT−3′) targeting the internal transcribed spacer rDNA sequence were designed with PRIMER3 software (http://bioinfo.ut.ee/primer3-0.4.0/ website).

A specificity test for the designed primer and probe was conducted by quantitative PCR assay with 12 strains (*Alexandrium minutum*, *Akashiwo sanguinea*, *Amphidinium carterae*, *Cochlodinium polykrikoides*, *Coolia malayensis*, *Dunaliella tertiolecta*, *Heterosigma akashiwo*, *O.* cf. *ovata*, *Prorocentrum micans*, *Prorocentrum minimum*, *Rhodomonas salina*, *Scrippsiella trochoidea*) (Figure 2). Thermal cycling was conducted under the following conditions: 95 °C for 3 min of initial denaturation then 40 cycles of amplification of 10 s at 95 °C and 30 s at 60 °C.

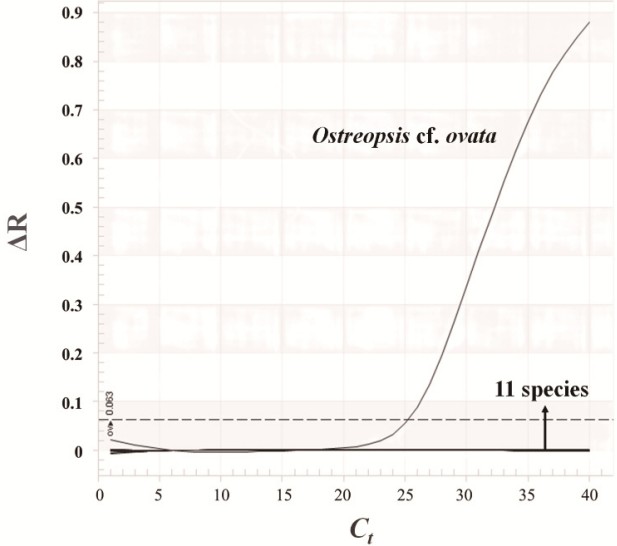

**Figure 2.** Test species specificity of *Ostreopsis* cf. *ovata* by quantitative polymerase chain reaction (PCR). A specificity test for the designed primer and probe was performed by quantitative PCR assay with 12 strains.

### 2.5. qPCR

The qPCR assay was conducted on a PCRmax Eco 48 real-time PCR system (PCR max, Stone, UK) using *O.* cf. *ovata* forward (5′-GGCCATTCCTAAGGACATCA−3′), *O.* cf. *ovata* reverse (5′-TGGCCATATAC AGCATGTTGAC−3′), and *O.* cf. *ovata* probe (5′-ATCATGCATTGTGTGAGTGTGTGATGT−3′) labeled at the 5′ and 3′ ends with the fluorescent dyes 6-FAM and BHQ−1. The final volume of the reaction mixtures for qPCR amplification was 20 μL and containing 10 μL of qPCRBIO Probe Mix No-ROX (2X) (PCR Biosystems, London, England), 1 μL of 10 μM each forward/reverse primer, 0.5 μL of 10 μM probe, 4.5 μL of UltraPure™ DNAse/RNAse-Free Distilled Water (Invitrogen, Carlsbad, CA, USA), and 3 μL of DNA template. The reagents were loaded into an Eco 48-well plate on the Eco 48 sample loading dock, and the plate was covered with an Eco 48 Adhesive Plate Seal to block contaminants such as powder, seam, or dust. After centrifuging the plate for 90 s, thermal cycling was carried out under the following conditions: 95 °C for 3 min of initial denaturation followed by 40 cycles of amplification for 10 s at 95 °C and 30 s at 60 °C. The standard curve was constructed by comparing the quantification cycle (Ct) value with the raw fluorescent signal measured by qPCR to known cell concentrations of *O.* cf. *ovata* (Figure 3). Approximately 30,000 *O.* cf. *ovata* cells cultured under laboratory conditions were filtered with Whatman (GF/C) filters, and gDNA eluted in a final volume of 100 μL for each sample was extracted using an AccuPrep Genomic DNA Extraction Kit (Bioneer, Seoul, Korea). A serial dilution to prepare samples containing 3000, 1000, 300, 100, 30, and 10 cells was performed by adding sterile distilled water (Invitrogen, Calsbad, USA), and the standard curve was determined by the qPCR method as described above. The Ct value for each field sample was also investigated under thermal cycling conditions, described above, and the relatively quantified cell concentrations of *O.* cf. *ovata* were calculated using the standard curve. A template containing sterile distilled water was used as a negative control, and one of the samples used for standard curve construction was measured as a positive control.

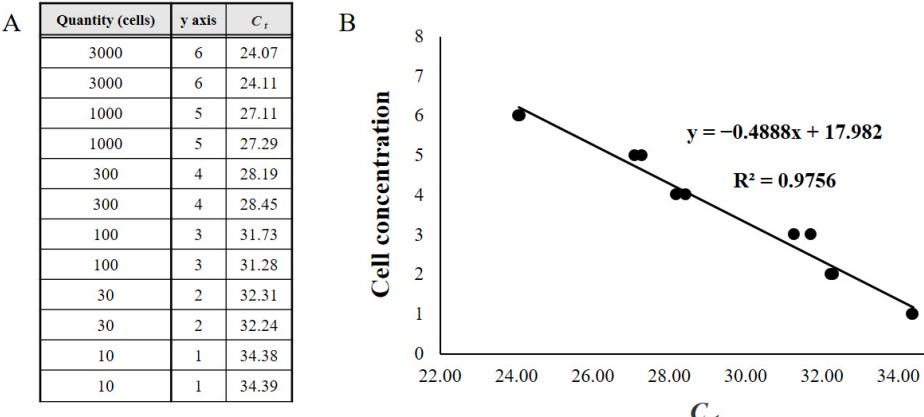

**Figure 3.** Standard curves for internal transcribed spacer rDNA sequence of *Ostreopsis* cf. *ovata*. (**A**) Six concentrations of 10–3000 cells were measured. (**B**) Standard curve generated by plotting the cell concentration versus threshold cycle ($C_t$). The slope was –0.488, correlation coefficient ($R^2$) was 0.976, and equation was $y = -0.4888x + 17.982$.

## 3. Results

### 3.1. Environmental Conditions

During the sampling period, the water temperature ranged from 11.3 °C in Gosan in January to 26.3 °C in Gosan in September (Table 2). The average temperatures in summer and autumn were similar (~ 22 °C) and were similar in winter and spring (15–16 °C). Salinity ranged from 22.7 to 36.1, with average value of 31.4. The salinity was relatively lower in summer in the harbor area (Jeju and Wimi), and it was affected by inland discharge after rainfall.

**Table 2.** Temperature and salinity measured during the sampling period at four sampling sites off Jeju coastal waters.

| Year | Month | Jeju (N) | | Gosan (W) | | Wimi (S) | | Seongsan (E) | |
|---|---|---|---|---|---|---|---|---|---|
| | | Temp. (°C) | Salinity | Temp. (°C) | Salinity | Temp. (°C) | Salinity | Temp. (°C) | Salinity |
| 2016 | July | 17.6 | 33.9 | 18.6 | 32.4 | 19.8 | 24.4 | 18.9 | 33.1 |
| | October | 19.5 | 26.4 | 22.5 | 34.5 | 22.3 | 28.8 | 21.7 | 32.7 |
| | December | 16.9 | 32.0 | 17.5 | 34.9 | 18.6 | 33.5 | 18.2 | 35.2 |
| 2017 | March | 15.5 | 28.1 | 15.9 | 36.1 | 15.8 | 33.1 | 14.4 | 35.0 |
| | July | 24.3 | 22.7 | 23.3 | 32.7 | 20.8 | 23.5 | 25.5 | 31.9 |
| | October | 20.0 | 27.1 | 21.9 | 33.1 | 21.9 | 30.4 | 21.2 | 32.8 |
| 2018 | January | 12.1 | 29.6 | 11.3 | 34.4 | 14.0 | 29.6 | 12.9 | 34.3 |
| | March | 16.0 | 28.7 | 17.0 | 34.3 | 16.2 | 28.2 | 14.6 | 34.1 |
| | July | 23.8 | 24.8 | 24.5 | 32.5 | 22.8 | 27.7 | 23.6 | 32.0 |
| | September | 24.4 | 33.2 | 26.3 | 32.6 | 24.2 | 26.4 | 24.3 | 32.4 |
| 2019 | January | 12.7 | 32.2 | 14.2 | 34.4 | 16.9 | 34.2 | 14.8 | 34.3 |

### 3.2. Collected Macroalgae Species and Attachment Rates

We collected 184 macroalgae samples from the four sampling sites off Jeju coastal waters from July 2016 to January 2019. In total, 26 species of macroalgae were collected, including green algae, red algae, and brown algae. *O.* cf. *ovata* was retrieved in 146 out of 184 samples, giving a 79.3% attachment rate; these values were 51% in red algae, 42% in brown algae, and 7% in green algae. The highest attachment rates for red algae were found for *Gelidium amansii* and *Corallina pilulifera*, brown algae for *Dictyopteris undulata* and *Ecklonia cava*, and green algae for *Ulva conglobate* and *Codium coactum* (Figure 4). The lowest attachment rate (33.3%) was observed in March 2017, and *O.* cf. *ovata* was found on all macroalgae samples in July 2018.

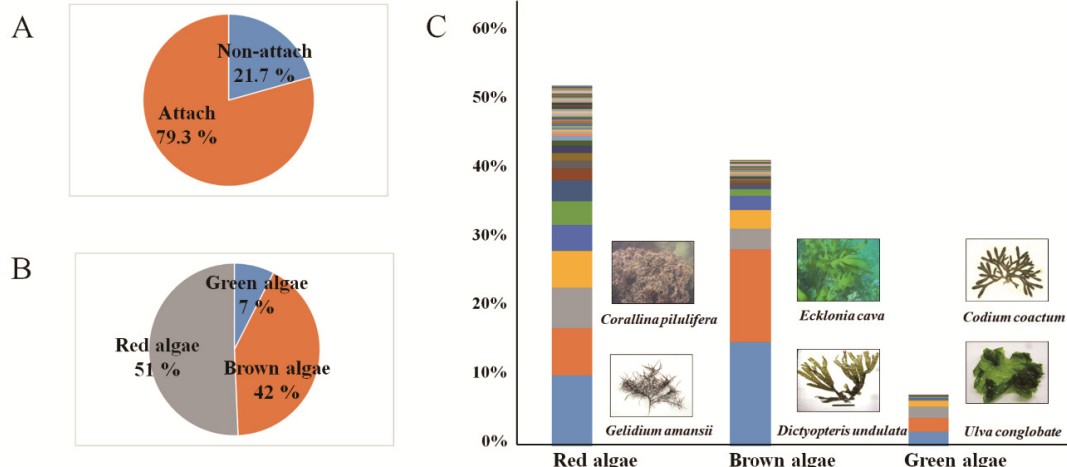

**Figure 4.** Attachment rates of *Ostreopsis* cf. *ovata* on collected macroalgae and their preference. (**A**) Attachment rates, *Ostreopsis* cf. was detected in 146 (79.3%) out of 184 macroalgae species. (**B**) Ratio of red (51%), brown (42%), and green (7%) algae among 146 macroalgae, (**C**) preferred macroalgae species.

### 3.3. Morphological Features of Ostreopsis cf. ovata

The cell shape of the *O.* cf. *ovata* had a typical oval to tear shape in the apical view and a thecal plate, which was dyed with calcofluor for easy identification an under epifluorescence microscope (Figure 5). The dorsoventral length and width of these cells were approximately 30–40 μm and 20–30 μm, respectively. The ratio of dorsoventral length to width was 1.2–1.5.

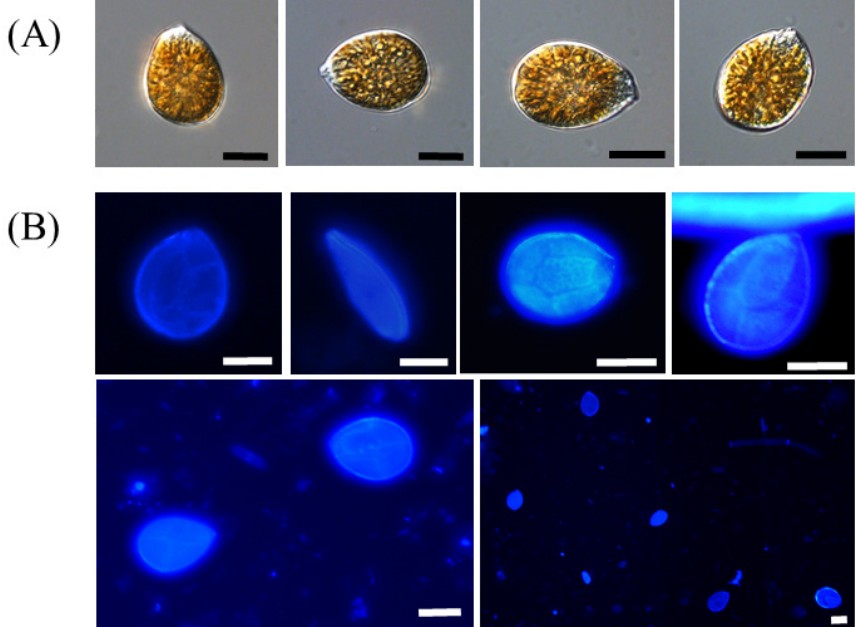

**Figure 5.** Photographs of *Ostreopsis* cf. *ovata* obtained by light microscopy (**A**) and calcofluor stained cells under epifluorescence microscopy (**B**) Scale bars = 20 μm.

### 3.4. Relationship between Microscopic Analysis and qPCR Results

To correctly identify and quantify *O.* cf. *ovata*, we combined the results obtained by microscopic analysis with those found by qPCR quantification. The data collected by microscopic analysis possibly included other *Ostreopsis* species, whereas qPCR method quantifies only the target species. A correspondence and high correlation was found between the results (Figure 6). The detection limit of microscopic analysis was low as 50 cells $g^{-1}$ fw macroalgae. In the range of low cell numbers, positive results were found by qPCR, even when no *Ostreopsis* cells were found by microscopy. Some data obtained by qPCR showed false positive or overestimated when no or low numbers of cells were detected by microscope. We excluded several data showing over 5000 cells $g^{-1}$ fw macroalgae obtained by qPCR as false positives after quick microscopic observation.

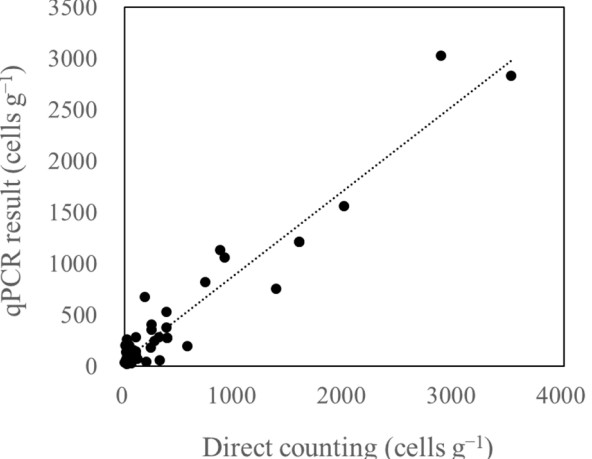

**Figure 6.** Relationship of abundance analyzed by microscopy and quantitative real-time polymerase chain reactions (qPCR). The slope was 0.8292, correlation coefficient ($R^2$) was 0.924, and the equation was $y = 0.8292x + 59.548$.

### 3.5. Temporal and Spatial Distributions of Ostreopsis cf. ovata

The abundance of the *O.* cf. *ovata* during the sampling period ranged from 4 cells $g^{-1}$ to 3204 cells $g^{-1}$ (Figure 7). In July 2016, the abundance at all four sites were very low, despite being in the summer season; however, in July 2017, the abundance increased to approximately 3000 cells $g^{-1}$ and then decreased during winter (December and January). The abundance increased again in summer, with the maximum abundance observed in September 2018 in the Jeju area (3204 cells $g^{-1}$). The abundance of *O.* cf. *ovata* fluctuated by season; it increased from summer to autumn and decreased from winter to spring. The abundance of *O.* cf. *ovata* during spring ranged from 119 to 319 cells $g^{-1}$, with 32–2825 cells $g^{-1}$ in summer, 4–3204 cells $g^{-1}$ in autumn, and 24–391 cells $g^{-1}$ in winter. The averaged abundance was lowest in winter (139.3 cells $g^{-1}$) and highest in autumn (704.4 cells $g^{-1}$), with 183.7 cells $g^{-1}$ in spring, and 462.2 cells $g^{-1}$ in summer. The abundance of *O.* cf. *ovata* increased after the water temperature reached 20 °C.

The highest abundance was observed at the northern coast of Jeju Island area (Figure 7). In this area, the highest abundance was recorded in September 2018 and ranged 32–3204 cells $g^{-1}$. The abundance ranged from 54 to 1.560 cells $g^{-1}$ in the western coastal area (Gosan), 24–2825 cells $g^{-1}$ in the eastern coastal area (Seongsan), and 4–504 cells $g^{-1}$ in the southern coastal area (Wimi). The average abundance in the Wimi area (southern coast) was lowest at 181.5 cells $g^{-1}$. The average abundances at Seongsan (east) and Gosan (west) were 402.8 and 481.4 cells $g^{-1}$, respectively, and was high at the northern coastal area, at 492.6 cells $g^{-1}$. The abundance of *O.* cf. *ovata* differed both temporally and spatially; it was high in the west and north in autumn, and east in summer. The southern coastal area (Wimi) showed low abundance throughout the year.

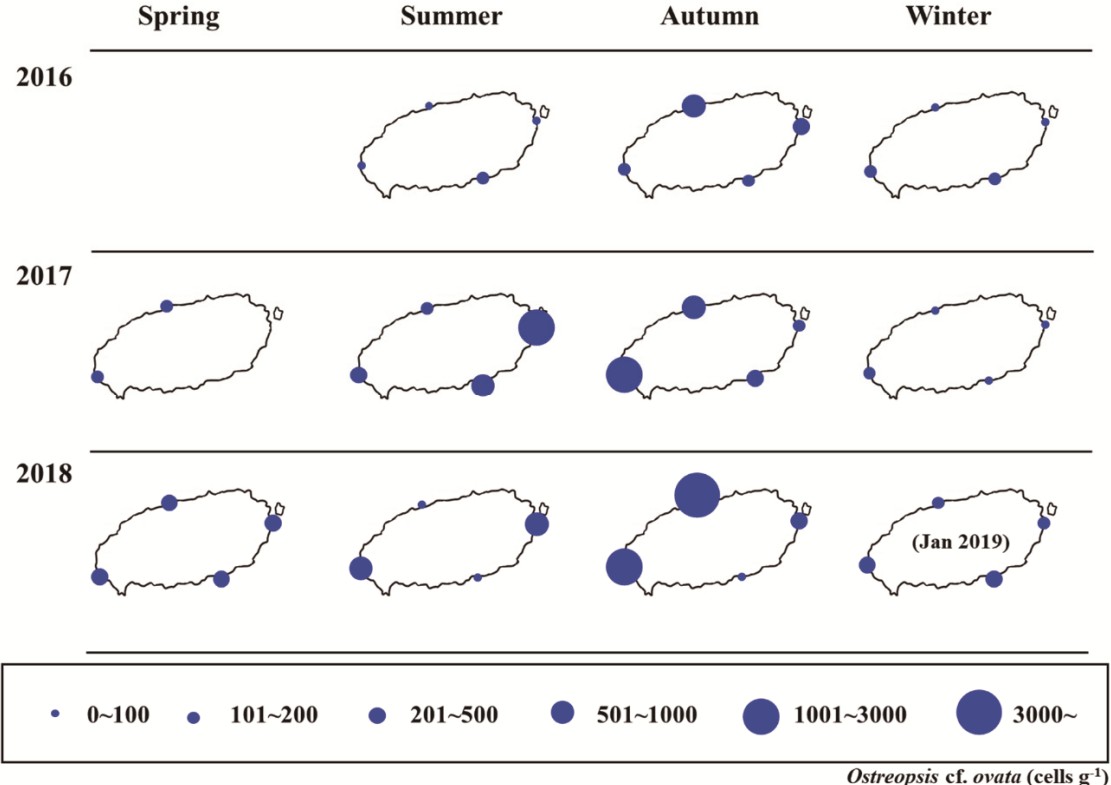

**Figure 7.** Spatio-temporal variations in the abundance (cells $g^{-1}$) of *Ostreopsis* cf. *ovata* at four sampling sites off Jeju coastal waters by season from July 2016 to January 2019.

## 4. Discussion

*Ostreopsis* species around Jeju Island has been reported since 2010 [44,50]. Moreover, Hwang et al. [13,51] reported the novel cytotoxic non-PLTX compounds Ostreol A and B in the strain isolated from Jeju Island, which have not both been reported in any other strain. Ostreol A showed very high toxicity in an in vitro cytotoxicity test against *Artemia*, but how it affects humans remains unclear.

We quantified *O.* cf. *ovata* over three years in Jeju coastal waters and determined the temporal and spatial distributions. We observed a high abundance in the northern and western coastal areas of Jeju Island, which has popular beaches and is frequently visited by tourists. We found that *O.* cf. *ovata* proliferated when the water temperature was >20 °C, when the beaches are open to tourists in summer. Mangialajo et al. [16] reported the proliferation of *O. ovata* along the urbanized coast of Italy in 2005, which caused human health problems with similar symptoms after exposure to marine aerosols (Table 2). At that time, the abundance of *O. ovata* was $2.5 \times 10^6$ cells $g^{-1}$ and approximately 200 received local first aid. In this research, the maximum abundance of *O.* cf. *ovata* was $3.2 \times 10^3$ cells $g^{-1}$, which is not as high as in the Mediterranean Sea, however *O.* cf. *ovata* from Jeju area contains another toxin that has not been reported in the strain from the Mediterranean Sea, the potential toxicity of which remains unclea

In this study, various macroalgae species were collected because macroalgae appear to be the preferred substrate for *O.* cf. *ovata*. Research by Tindall and Morton showed that the maximum density of epiphytic species attached to macroalgae was generally $10^2$–$10^4$ cells $g^{-1}$ [52], and these species tended to attach to Rhodophyta and Phaeophyta at high densities, particularly to *Halopteris scoparia*, which showed the highest recorded density ($5.9$–$10^5$ cells $g^{-1}$) [31]. Our results showed similar tendency to the results of these studies. The three-dimensional flexibility and high surface area of the macroalgae can explain the preference of *O.* cf. *ovata* for this substrate rather than coral or sand. Additionally, the detection of epiphytic dinoflagellates in the sampling area follows a clear seasonal pattern [53].

The abundance of *O.* cf. *ovata* widely varies by season, increasing from summer to autumn and then decreasing from winter to spring, revealing a relationship with water temperature. The average water temperature in Jeju coastal waters in summer and autumn was 22 °C and in winter and spring was 15 °C. In our laboratory test, *O.* cf. *ovata* isolated from Jeju waters showed optimal growth at 15–20 °C and did not survive below 10 °C (unpublished data). The optimal salinity of *O.* cf. *ovata* ranged from 30–40, and the salinity range of the study areas was 22.7–36.1, indicating that when the salinity is low, the growth of *O.* cf. *ovata* could depressed. In the present study, *O.* cf. *ovata* was abundant when the water temperature was 22–25 °C and salinity was 31–34 (Figure 8). These conditions were the same as the optimal conditions determined in the laboratory growth experiment (unpublished data). In Jeju Island, submarine groundwater discharge in a coastal area was reported [54], which affected the salinity and inorganic nutrient concentration. Jauzein et al. [55] reported that *O.* cf. *ovata* has a high uptake ability and availability of nitrate when the nitrate concentration was high. In addition, Lee and Kim [56] suggested that submarine groundwater discharge can lead to the blooming of monospecific dinoflagellates by increasing nitrate concentrations. Therefore, submarine groundwater discharge in the Jeju coastal area, which has a high nutrient concentration, may promote the proliferation of *O.* cf. *ovata* but may also possibly play an inhibitory role by lowering salinity.

In conclusion, we found that *O.* cf. *ovata* was distributed around the Jeju coastal area and abundant when the water temperature was over 20 °C. This species was abundant in the northern and western coastal areas and survived over the winter in the Jeju coastal waters. The environmental conditions of the Jeju coastal waters may promote the growth of *O.* cf. *ovata*. Further monitoring and research are needed to evaluate the proliferation of *O.* cf. *ovata*, which contains a novel toxin with unidentified effects on humans.

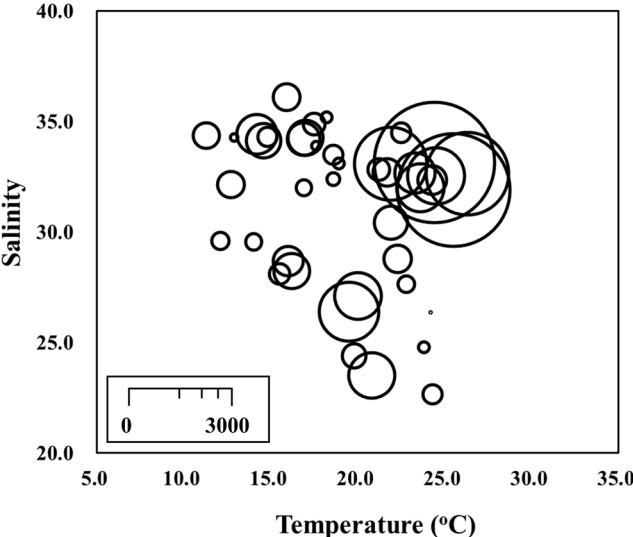

**Figure 8.** Abundance (cells g$^{-1}$) of *Ostreopsis* cf. *ovata* in Jeju coastal waters during the study period as a function of water temperature and salinity. Circle size: number of cells (cells g$^{-1}$).

**Author Contributions:** Data curation, J.P., J.H. and E.Y.Y.; Formal analysis, J.-H.H.; Funding acquisition, J.P.; Investigation, J.H.; Methodology, J.-H.H.; Project administration, J.P.; Writing—original draft, J.P. and E.Y.Y.; Writing—review and editing, J.P., J.H. and E.Y.Y. All authors have read and agreed to the published version of the manuscript.

**Funding:** This research was a part of the project titled "Improvement of management strategies on marine disturbing and harmful organisms (No. 20190518)" funded by the Ministry of Oceans and Fisheries, Korea, and supported by the National Research Foundation of Korea (NRF) grant funded by the Korea government (MSIT) (NRF−2018R1D1A1B07047821).

**Conflicts of Interest:** The authors declare no conflict of interest.

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
