# Peer review of "Temporal and Spatial Distribution of the Toxic Epiphytic Dinoflagellate Ostreopsis cf. ovata in the Coastal Waters off Jeju Island, Korea"

_sustainability, doi:10.3390/su12145864_

Round 1

Reviewer 1 Report

Evaluation of the manuscript: Temporal and spatial distribution of the toxic epiphytic dinoflagellate Ostreopsis cf. ovata in the coastal waters off Jeju Island, Korea

General considerations

This paper refers about several years of research on the presence and the distribution of O. ovata in coastal Kirean waters. The paper appears well written and the topic is suitable for the publication in MDPI Sustainability. This is quite an actual topic given its implications for human activities and its relationships with both benthic and planktonic communities. In several parts the manuscrips is wrongly written, and it needs deep revision by a mother-language author or an English speaking biologist. I added some main suggestions in the annotated manuscript, but I strongly suggest a deep revision of the text both for the contents and the language correctness because in several points it appears not comprehensible. In general I found the methods appropriate and the results worth to be published, especially because they show spatial and temporal trends that were unpublished in this shape. However, the manuscript needs a deep re-writing to reach the level needed for a scientific publication because in several parts it is totally incomprehensible. I indicated the main flaws in the next sections and I strongly encourage the authors to re-write these interesting findings and re-submit after major revision.

Introduction

Some sentences as “The genus Ostreopsis is a toxic dynoflagellate…” are misleading and, as far as I understand, should mean “The genus Ostreopsis comprises toxic marine dynoflagellates..."”. Besides this, the introduction is very poor and does not cover the various aspects of O. ovata in various areas of the world. For sure the authors could do better, by adding at least a page of introductive concepts explaining the actual relationships found between the presence of these toxic organisms and other organisms interacting with them, as well as some information about their geographical distribution, the frequency of occurrence in coastal areas, and the actual damages produced to human health. Also the possibility to live both in the water column and in the benthos is essential to understand their ecology, as well as their relationships with competing organisms, as cyanobacteria, both in benthic and in planktonic environments. These introductory concepts could be essential to justify their research.

Material and methods

The methods appear quite accurate as for the molecular investigations but some details on the sampling plan could help. For example I cannot understand well how replicates were established in each place and how the number of collections was set (based on minimum collection strategy? Patchiness? Or what?). On the whole however the methods appear appropriate and useful to reach the results.

Results

They appear quite interesting also for the innovative representation of spatial and temporal trends. However, in several parts they are totally incomprehensible. For example, the units used to indicate abundances are quite smoky in the first part of results, then they return to normal values (e.g. nr cells per gram, although this measure was never justified in the methods) in the second part of results.

Discussion

It is enough good but (besides language) I am sure the authors could reach more exciting conclusions using the good data sets collected. I suggest to make an effort for comparing their results with those of previous authors and identify interesting relationships within the abundance of the protozoans with the species of algae, their characters, and the typical ecosystems where they are normally fitting best, eventually in parallel with the trends of chemical and physical measures.

Figures

Some figures appear clear. But some are totally incomprehensible. For example in figure 4, I cannot understand what really the numbers under the X axis mean. Are those years? From 1607 to 1901? Or what? None is indicated in the caption nor in the text

Figure 7 should be introduced in the results and then discussed, not appearing in the discussion without any previous introduction.

Tables

Table 2 should be reported in the introduction and there discussed. Not needed in the results, since it is not a finding of this study

Of course table 1 should be presented BEFORE table 2! Also this could find good positioning in the results, not in the discussion.

Literature

I did not check all the correspondences of literature citations since this can be done on the final version of the manuscript.

Author Response

Reviewer 1.

General considerations

This paper refers about several years of research on the presence and the distribution of O. ovata in coastal Korea waters. The paper appears well written and the topic is suitable for the publication in MDPI Sustainability. This is quite an actual topic given its implications for human activities and its relationships with both benthic and planktonic communities. In several parts the manuscript is wrongly written, and it needs deep revision by a mother-language author or an English speak biologist. I added some main suggestions in the annotated manuscript, but I strongly suggest a deep revision of the text both for the contents and the language correctness because in several points it appears not comprehensible. In general, I found the methods appropriate and the results worth to be published, especially because they show spatial and temporal trends that were unpublished in this shape. However, the manuscript needs a deep re-writing to reach the level needed for a scientific publication because in several parts it is totally incomprehensible. I indicated the main flaws in the next sections and I strongly encourage the authors to re-write these interesting findings and re-submit after major revision.

Introduction

Some sentences as “The genus Ostreopsis is a toxic dynoflagellate…” are misleading and, as far as I understand, should mean “The genus Ostreopsis comprises toxic marine dinoflagellates..."”. Besides this, the introduction is very poor and does not cover the various aspects of O. ovata in various areas of the world. For sure the authors could do better, by adding at least a page of introductive concepts explaining the actual relationships found between the presence of these toxic organisms and other organisms interacting with them, as well as some information about their geographical distribution, the frequency of occurrence in coastal areas, and the actual damages produced to human health. Also the possibility to live both in the water column and in the benthos is essential to understand their ecology, as well as their relationships with competing organisms, as cyanobacteria, both in benthic and in planktonic environments. These introductory concepts could be essential to justify their research.

We corrected it in the text file as suggested.

Line 30 The genus Ostreopsis contains toxic marine dinoflagellates,

Line 37 Recently, the global warming was shown to affect the marine ecosystem structure and the outbreaks of harmful algal blooms and PLTX caused by Ostreopsis species

Line 39~58 Add sentence

Especially Ostreopsis cf. ovata produces a wide range of palytoxin-like compounds and ovatoxins (ovatoxin, a~f), [10-12] among which ovatoxin is a non-palytoxin compounds recently reported as marine toxin affecting shrimp [13].

In previously decades, Ostreopsis blooms were common in temperate water, mostly in the Mediterranean Sea during summer to autumn and other temperate areas (Table 1). It has also been reported that the winter water temperature is below 0°C, and their distribution range of this genus is expanding worldwide [14].

Blooms of Ostreopsis species caused serious human health problems via inhalation of sea water droplets containing Ostreopsis cells or aerosolized toxins [6, 15, 16]. Ostreopsis can create and proliferate mucus floating clusters. These clusters float on the water surface and can be aerosolized by the release of toxic cells and related toxins through cell lysis and from sea spray [6, 17-19]. In addition, as a primary producer, these organisms plays an important role in the marine food chain and are a feed source for large protozoa or shellfish through filter-feeding. Consumption of Ostreopsis induces paralytic shellfish poisoning, diarrhea shellfish poisoning and neurotoxic shellfish poisoning [11]. Thus, the dinoflagellate toxin and its analogs can be transfer to humans. Respiratory symptoms from sea spray were first reported in 2003 Italy. The abundance of Ostreopsis was over 2.5 x 106 cells g-1 fw of macrophyte, and high cell concentration were detected in the water column. Two hundred people exposed to marine aerosols received local first aid treatment, and among them 20 people required extended hospitalization

Material and methods

The methods appear quite accurate as for the molecular investigations but some details on the sampling plan could help. For example I cannot understand well how replicates were established in each place and how the number of collections was set (based on minimum collection strategy? Patchiness? Or what?). On the whole however the methods appear appropriate and useful to reach the results.

We corrected it in the text file as suggested.

Line 80 Macroalgae samples were collected by scuba divers from July 2016 to January 2019 within 10 meters of water depth to collect the macroalgae attached O. cf. ovata. Four different macroalgae species were collected at the four sampling sites (Jeju-north, Gosan-west, Wimi-south, Seongsan-east) off Jeju Island, (Fig.1).

Results

They appear quite interesting also for the innovative representation of spatial and temporal trends. However, in several parts they are totally incomprehensible. For example, the units used to indicate abundances are quite smoky in the first part of results, then they return to normal values (e.g. nr cells per gram, although this measure was never justified in the methods) in the second part of results.

We corrected it in the text file as suggested.

In the first result, the figure 4 was modified, and in the second result, the abundances of O. cf. ovata was expressed in cells per gram.

Discussion

It is enough good but (besides language) I am sure the authors could reach more exciting conclusions using the good data sets collected. I suggest to make an effort for comparing their results with those of previous authors and identify interesting relationships within the abundance of the protozoans with the species of algae, their characters, and the typical ecosystems where they are normally fitting best, eventually in parallel with the trends of chemical and physical measures.

We corrected it in the text file as suggested.

Add sentence

Line 302 In this study, various macroalgae species were collected by selecting macroalgae as the preferred substrate for O. cf ovata. A study by Tindall & Morton [1998] showed that the maximum density of epiphytic species attached to macroalgae was generally 102–104 cells g-1, but these species tended to attach to Rhodophyta and Phaeophyta at high densities, particularly in Halopteris scoparia, which showed the highest recorded density (5.9-105 cells g-1) [35]. In addition, the higher frequency in macroalgae than on other attachment substrate such as coral and sand, it is known as a morphological factor with three-dimensional flexibility and high-surface area. Additionally, the detection of epiphytic dinoflagellates in the sampling area follow a clear seasonal pattern. Changes in temperature, illuminance and tidal currents with seasonal changes are a highly correlated with the concentrations of epiphytic dinoflagellates [36].

Figures

Some figures appear clear. But some are totally incomprehensible. For example in figure 4, I cannot understand what really the numbers under the X axis mean. Are those years? From 1607 to 1901? Or what? None is indicated in the caption nor in the text

We corrected it in the figure 7 as suggested.

From 1607 to 1901 à from Jul-16 to Jan-19

Figure 7 should be introduced in the results and then discussed, not appearing in the discussion without any previous introduction.

We deleted the figure 7 that was not mentioned in the introduction, as suggested

Tables

Table 2 should be reported in the introduction and there discussed. Not needed in the results, since it is not a finding of this study

Of course table 1 should be presented BEFORE table 2! Also this could find good positioning in the results, not in the discussion.

We changed the table order as suggested. In addition, a description of the table 1 has been added to the introduction section.

Line 40 In previously decades, Ostreopsis blooms were common in temperate water, mostly in the Mediterranean Sea during summer to autumn and other temperate areas (Table 1).

Literature

I did not check all the correspondences of literature citations since this can be done on the final version of the manuscript.

We checked references and citations as suggested.

Line 20 Not comprehensible, please reformulate this quantity

à Line 19 4-3,204 cells g-1

Line 29 To be totally rephrased as: The genus ostreopsis comprises toxic marine dynoflagellates

à Line 30 The genus Ostreopsis contains toxic epiphytic marine dinoflagellates

Line 34 could serious problems, please explain the concept. Apparently a verb is missing

à Line 37 Recently, the global warming was shown to affect the marine ecosystem structure and the outbreaks of harmful algal blooms and PLTX caused by Ostreopsis species

Line 39 Appearance is definitely not the right term here. The introduction? The life?, the evolution?

à Line 65 Jeju Island is located in temperate region, but its coastal waters are affected by the Tsushima Warm Current, which introduces tropical fish and invertebrates

Line 89 How is possible to transfer a single cell into 96 wells? Please explain better this concept. Maybe you produced monocional cultures in 96 replicate wells?

à Line 117 To establish a single cell strain of O. cf. ovata, the live sample transferred to the laboratory was placed in a 6-well plate and single cell isolation was performed under a dissecting microscope. Clonal cultures of O. cf. ovata were established by 2 serial single-cell isolations. After sufficient growth, the cells were transferred into a 30-mL flask.

Line 145 This sentence is totally lacking any common sense. Please rephrase

à Line 207 O. cf. ovata was found in 146 samples, giving a 79.3% attachment rate; these values were 51% in red algae, 42% in brown algae and 7% in green algae. The highest attachment rates for red algae were found for Gelidium amansii and Corallina pilulifera, brown algae for Dictyopteris undulata and Ecklonia cava, and green algae for Ulva conglobate and Codium coactum (Fig. 4). The lowest attachment rate (33.3%) was observed in March 2017, and O. cf. ovata was found in all macroalgae samples in July 2018.

Line 167 was varied is definitely not an English way to say, probably you meant” varied according to seasons.

We removed the sentence according to suggestions from other reviewers

Reviewer 2 Report

Park et al. have investigated the temporal and spatial distribution of the toxic epiphytic dinoflagellate Ostreopsis cf. ovata in the coastal waters off Jeju Island, Korea, through a long-term investigation. Because there is a lot of research data in MS, it is recommended to re-examine after revision. The points to focus on when modifying the Manuscript are as follows.

  • In referring to the cell density of Ostreopsis, if both the results of microscopic and QRT PCR are used, the correlation between the cell density calculated by both methods should be presented; Otherwise, mention only the microscope population and remove the part of QRT PCR from the methodology.
  • Discuss more intensely the relationship between marcroalgae and target Ostreopsis.
  • Add discussion of water temperature and salinity (as indicated in Table 2); In particular, low salinity was frequently observed in the survey area, so focus on this.
  • Describe the formula for calculating the growth rate from materials and methods, species isolation and maintenance, and experimental methods.
  • Nutrients, pH, and DO are not mentioned in the discussion, so if you do not discuss them further, you should delete them; In particular, the concentration of nutrients is quite high. Have you ever analyzed using seaweed extraction water?
  • The rest of the details should be modified with reference to the attached PDF file.

Author Response

Reviewer 2

In referring to the cell density of Ostreopsis, if both the results of microscopic and QRT PCR are used, the correlation between the cell density calculated by both methods should be presented; Otherwise, mention only the microscope population and remove the part of QRT PCR from the methodology.

We added a figure 6 of the correlation of microscopic and rt-PCR with cell density. Also, rt-PCR standard curve has been added as figure 3.

Discuss more intensely the relationship between marcroalgae and target Ostreopsis.

As suggested, we added the relationship between marcroalgae and target Ostreopsis to the discuss section.

Line 266 In this study, various macroalgae species were collected by selecting macroalgae as the preferred substrate for O. cf ovata. A study by Tindall & Morton [1998] showed that the maximum density of epiphytic species attached to macroalgae was generally 102–104 cells g-1, but these species tended to attach to Rhodophyta and Phaeophyta at high densities, particularly in Halopteris scoparia, which showed the highest recorded density (5.9-105 cells g-1) [35].

Add discussion of water temperature and salinity (as indicated in Table 2); In particular, low salinity was frequently observed in the survey area, so focus on this.

We corrected it in the text file as suggested.

Line 40 In previously decades, Ostreopsis blooms were common in temperate water, mostly in the Mediterranean Sea during summer to autumn and other temperate areas (Table 1). It has also been reported that the winter water temperature is below 0°C, and their distribution range of this genus is expanding worldwide [14].

Line 270 In addition, the higher frequency in macroalgae than on other attachment substrate such as coral and sand, it is known as a morphological factor with three-dimensional flexibility and high-surface area. Additionally, the detection of epiphytic dinoflagellates in the sampling area follow a clear seasonal pattern. Changes in temperature, illuminance and tidal currents with seasonal changes are a highly correlated with the concentrations of epiphytic dinoflagellates [36].

Describe the formula for calculating the growth rate from materials and methods, species isolation and maintenance, and experimental methods.

We removed the unexplained growth rate figure 7.

Nutrients, pH, and DO are not mentioned in the discussion, so if you do not discuss them further, you should delete them; In particular, the concentration of nutrients is quite high. Have you ever analyzed using seaweed extraction water?

We deleted nutrient data that was not discussed as suggested.

The rest of the details should be modified with reference to the attached PDF file.

Line 16 Please remove this sentence or show the reliable data. Please discuss the relationship between occurrences of ovate and macroalgae, if you want to add this in abstract

We corrected it in the text file as suggested.

Line 199 The highest attachment rates for red algae were found for Gelidium amansii and Corallina pilulifera, brown algae for Dictyopteris undulata and Ecklonia cava, and green algae for Ulva conglobate and Codium coactum (Fig. 4). The lowest attachment rate (33.3%) was observed in March 2017, and O. cf. ovata was found in all macroalgae samples in July,

Add figure 4

Line 22 The growth condition was not examined in this study, although you showed the related figure. If you want to add this, please specify the methods and results.

We deleted growth condition as suggested.

Line 25 Four Ostreopsis species (O. cf. ovata, O. lenticularis, O. mascarenensis, and O. siamensis) à Four Ostreopsis species, O. cf. ovata, O. lenticularis, O. mascarenensis, and O. siamensis, produce PLTX-like compounds

Line 27 The impact of global warmingà Recently, the global warming was shown to affect the

Line 65 area à coastal waters are

Line 66 causingà which introduces

Line 70 there is a dearth of studies on the temporal and spatial distribution of Ostreopsis cf. ovata off Jeju coastalà However, the temporal and spatial distribution of Ostreopsis cf. ovata off Jeju coastal waters is not well-understood.

Line 72 explore à investigated

Line 74 samples fromà samples collected from

Line 79 Collecting samples and treatmentà Sample collection and treatment

Line 83 Environmental factors, such as water temperature, salinity, Ph and dissolved oxygen, were measured in the surface water using YSI pro (? Company, model name…. )à Environmental factors, such as water temperature, salinity, pH, and DO, were measured in the surface water using a YSI Professional plus (Xylem, Inc., Rye Brook, USA)

Line 85 The macroalgaeà The collected macroalgae

Line 89 shakenà which as filled with filtered seawater, and then shaken vigorously

Line 94 establishà as a live sample for establishing strains

Line 95 filtrate à surface sea water filtered

Line 106 2.2 Fluorescence microscopic counting with Calcofluor stainingà Fluorescence microscopy

Line 111 The photographs were takenà The cells were photographed with a DP73 digital camera system

Line 116 2.3 Strain setupà Culture, DNA extraction and PCR amplification

Line 119 After growth of the strain was confirmed, it wasà After sufficient growth, the cells were transferred into a 30-mL flask

Line 134 2.4. Design of species-specific primer

We added the resulting figure 2 as suggested.

Line 152 2.5. Quantitative PCR (qPCR)

We added the resulting figure 3 that rt-PCR standard curve as suggested.

Line 203 3.1. Collected macroalgae species and attachment rates

à We removed the previous figure 2 and added a new figure 4.

Line 197 The salinity ranged from 22.7–36.1, with a 31.4 average. The nitrate concentration ranged from 2.5–96.2 μM, with a 24.8 μM average. The nitrate concentration was varied by season, with the highest concentration was in summer (40.5 μM) and the lowest in spring (15.2 μM). The phosphate concentration ranged from 0.2–1.7 μM, and the average was 0.6 μM. The phosphate concentration showed no seasonal differences.

à The salinity was relatively lower in summer in the harbor area (Jeju and Wimi), it was affected by inland discharge after rainfall.

Line 225 3.3 temporal distribution, 3.4 spatial distribution à 3.5. Temporal and spatial distributions of Ostreopsis cf. ovata

Line 286 Ostreopsis species are known to produce PLTX-like compounds and cause food poisoning by bioaccumulation in the marine ecosystem or sometimes causing direct illness in humans [16, 17]. It is known to be distributed worldwide and their presence around Jeju Island has been reported since 2010 [18, 19]. Moreover, Hwang et al.[20, 21] reported the novel cytotoxic non-PLTX compounds Ostreol A and B from the strain isolated from Jeju Island, which have not both been reported in any other strain. Ostreol A shows very high toxicity in the vitro cytotoxicity test against Artemia, but how it affects humans remains unclear.

à Ostreopsis species around Jeju Island has been reported since 2010 [32, 33]. Moreover, Hwang et al. [13, 34] reported the novel cytotoxic non-PLTX compounds Ostreol A and B in the strain isolated from Jeju Island, which have not both been reported in any other strain. Ostreol A showed very high toxicity in an in vitro cytotoxicity test against Artemia, but how it affects humans remains unclear.

Line 278 the inorganic nitrate concentration in the studying area was very high, found to be up to 100 μM (Table 1), and average nitrate concentration was over 40 μM in summer.

We deleted the sentence as suggested.

Line 311 Figure 7. The specific growth rate (d-1) of O. cf. ovata under various temperatures (A) and salinities(B). The error bar represents the standard deviation from the mean of triplicate data (n = 3)(unpublished data).

We deleted figure 7 as suggested.

Round 2

Reviewer 1 Report

Evaluation of the manuscript:

Temporal and spatial distribution of the toxic epiphytic dinoflagellate Ostreopsis cf. ovata in the coastal waters off Jeju Island, Korea.

Overview

This manuscript aims at detecting spatial and temporal distribution of a toxic alga in the benthos of Jeju Island. Ostreopsis is a puzzling issue because its ecology and the relationships with other competitors for space and resources are still underexplored. Thus another paper on various species of Ostreopsis performed in this Korean environment may be important, after the first reports by Kim (2009) and Baek (2012). The topic is interesting and the findings are worth to be published. English language needs deep revision by a mother-language person because it contains several bad errors. I do not understand why some words are indicated in red in the manuscript. Maybe this is an editorial rule I ignore, however, or they may derive by previous revisions.

Abstract

It is fine with the topics but contains various small errors. I have highlighted some in the annotated pdf.

Introduction

Is almost complete and well structured. However, also here various issues with the English language were found. Some of them have been highlighted in the annotated pdf as an example, but I suggest a deep revision by a mother language colleague. In addition, at line 68 it has been stated that O. ovate is a benthic dynoflagellate, but this species may be present also in the plankton (see for example Ocurrence of Ostreopsis in two temperate coastal bays (SW iberia): Insights from the plankton. Santos, Mariana ; Oliveira, Paulo B ; Moita, Maria Teresa ; David, Helena ; Caeiro, Maria Filomena ; Zingone, Adriana ; Amorim, Ana ; Silva, Alexandra. Harmful Algae, 2019-06, Vol.86, p.20-36). This should be revised. As well, please check if all the occurrences reported in table 1 are referred to benthos and, if also plankton is considered by some papers, indicate it extensively to show the actual patterns of seasonal and spatial distribution.

Methods

I will be repetitive but also methods require extensive linguistic revision: several sentences are simply impossible to understand. I indicated some key points in the annotated pdf

Results

Some terms are probably wrong as indicated in the pdf. Some more details would be helpful in the description of the ecological conditions of sites. Since an evident spatio-temporal trend is observed among stations, I also recommend to test some possible relationships with temperature and/or salinity. For example you could attempt a regression between temperatures in all sites and abundance in all sites (eventually using 4 different colors for dots coming from different sites). And you could do the same for the couple salinity vs abundance. In case of positive relationships, you could have additional records to discuss.

Discussion

The concept exposed at lines 283-284 could be circumstantiated and demonstrated if a regression is reported in the results, as above indicated. Also discussion needs a deep linguistic revision.

Conclusions

I would not add a section just to repeat some basic concepts of the manuscript. In case incorporate at the end of the discussion, or just discharge those sentences.

Figure 4. I cannot understand A: attachment rates. What this refers to? Use a specific description in the caption and be more specific in all figure captions.

Figure 9. You do not indicate what the circles refer to: number of cells? Be explicit and indicate also what the scale bar is referring to.

Author Response

Abstract

It is fine with the topics but contains various small errors. I have highlighted some in the annotated pdf.

-> We corrected it in the text file as suggested.

Line 16 The water temperature of Jeju Island in winter remained higher than 15°C, and this species could be overwintering in the Jeju waters.

Line 17 further

Introduction

Is almost complete and well structured. However, also here various issues with the English language were found. Some of them have been highlighted in the annotated pdf as an example, but I suggest a deep revision by a mother language colleague. In addition, at line 68 it has been stated that O. ovate is a benthic dynoflagellate, but this species may be present also in the plankton (see for example Ocurrence of Ostreopsis in two temperate coastal bays (SW iberia): Insights from the plankton. Santos, Mariana ; Oliveira, Paulo B ; Moita, Maria Teresa ; David, Helena ; Caeiro, Maria Filomena ; Zingone, Adriana ; Amorim, Ana ; Silva, Alexandra. Harmful Algae, 2019-06, Vol.86, p.20-36). This should be revised. As well, please check if all the occurrences reported in table 1 are referred to benthos and, if also plankton is considered by some papers, indicate it extensively to show the actual patterns of seasonal and spatial distribution.

-> We corrected it in the text file as suggested. Also, we got English correction by mother language expert three times.

Line 25 have been expanding

Line 29 (rephrase) Blooms of PLTX-producing algae cause serious problems in aquatic ecosystem and fisheries and are dangerous to human health [7, 8]. Recently, the global warming has been shown to affect the marine ecosystem structure and promote outbreaks of harmful algal blooms with consequence production of PLTX caused by Ostreopsis species [9].

Line 41 (rephrase) These clusters float on the water surface, and toxins released from the cells through cell lysis can be aerosolized sea spray [6, 17-19].

Line 42 In addition, these organisms

Line 45 can be transferred

Line 49 (add sentence) Also, the distribution of Ostreopsis species from 1985 to 2017 has been reported to occur in more two million cells g–1 in the Mediterranean at temperature ranged > 20℃ and salinity ranged > 30 (Table 1).

Line 63 to investigate

Line 70 benthic dinoflagellates

Line 71 damage are based on

Methods

I will be repetitive but also methods require extensive linguistic revision: several sentences are simply impossible to understand. I indicated some key points in the annotated pdf

-> We corrected it in the text file as suggested.

Line 81 deleted sentence

Line 84 were, add reference

Line 85 The collected macroalgae were

Line 100 (add sentence) The fixed 300 ml of seawater was placed over a day to allow the cells to sink, and the supernatant was removed to adjust the final volume to 50-60 ml. In addition, the frozen macroalgae were weighed after removed seawater

Line 177 11.3℃ in Gosan in January to 26.3℃ in Gosan in September (Table 2). Delete next sentence (repeated)

Line 188 O. cf. ovata was retrieved in 146 out of 184 samples

Line 192 on

Results

Some terms are probably wrong as indicated in the pdf. Some more details would be helpful in the description of the ecological conditions of sites. Since an evident spatio-temporal trend is observed among stations, I also recommend to test some possible relationships with temperature and/or salinity. For example you could attempt a regression between temperatures in all sites and abundance in all sites (eventually using 4 different colors for dots coming from different sites). And you could do the same for the couple salinity vs abundance. In case of positive relationships, you could have additional records to discuss.

We tried to find the differences between sites by analyzing relationships of abundance and temperature or salinity, but there were no trends or clear difference among the 4 sites. We thought that the difference among the sites was caused by tidal effects or regional currents and differences in geography by sites, but we do not mention them in this research due to the lack of evidence or data.

Discussion

The concept exposed at lines 283-284 could be circumstantiated and demonstrated if a regression is reported in the results, as above indicated. Also discussion needs a deep linguistic revision.

-> We corrected it in the text file as suggested.

Line 270 The abundance of O. cf. ovata widely varies by season, increasing from summer to autumn and then decreasing from winter to spring, revealing a relationship with water temperature.

Conclusions

I would not add a section just to repeat some basic concepts of the manuscript. In case incorporate at the end of the discussion, or just discharge those sentences.

-> We corrected it in the text file as suggested. Incorporate at the end of the discussion

Line 287 In conclusion, we found that O. cf. ovata was distributed around the Jeju coastal area and abundant when the water temperature was over 20℃.

Figure 4. I cannot understand A: attachment rates. What this refers to? Use a specific description in the caption and be more specific in all figure captions.

-> We corrected it in the text file as suggested.

Figure 4. Attachment rates of Ostreopsis cf. ovata on collected macroalgae and their preference. A. Attachment rates, Ostreopsis cf. was detected in 146 (79.3%) out of 184 macroalgae species B. ratio of red (51%), brown (42%) and green (7%) algae among 146 macroalgae, C. preferred macroalgae species

Figure 9. You do not indicate what the circles refer to: number of cells? Be explicit and indicate also what the scale bar is referring to.

-> We corrected it in the text file as suggested.

Figure 9. Abundance (cells g-1) of Ostreopsis cf. ovata in Jeju coastal waters during the study period as a function of water temperature and salinity. Circle size: number of cells (cells g-1)

Reviewer 2 Report

Park et al. have investigated the temporal and spatial distribution of the toxic epiphytic dinoflagellate Ostreopsis cf. ovata in the coastal waters off Jeju Island, Korea, through a long-term investigation. Although MS has been revised and improved in many ways, I think it should be published after further minor revision.

Introduction

Line 61 It feels sudden and needs to be modified more contextually and smoothly.

Line 65-66 This MS does not mention environmental factors other than water temperature and salt, so please delete the contents of nutrients and DO.

Materials and Methods

Line 83 As in the introduction, delete pH and DO not mentioned in MS.

Line 92-94 Delete Method for Nutrient.

Line 99-101 Need to be clearly rewritten.

Line 112 Delete unnecessary characters in PDF file.

Line 123-125 The target organism is an epiphytic organism, but did you just use seawater instead of shaking the seaweed for analysis? If so, there is a problem principally in the method.

Line 160 Delete unnecessary characters in PDF file.

Results

Line 199 Delete unnecessary characters in PDF file.

Line 244 and Line 257 Figures 7 and 8 appear to contain the same information. Either one needs to be deleted, and it is recommended to delete Figure 7.

Discussion

Line 280-282 It is a sudden expression that is not natural in context. Insufficient to account for seasonal changes in target organisms through this reference.

Line 297 In Figure 9, it does not appear to be inhibited by lower salinity. The expression needs to be somewhat relaxed.

Conclusion

Line 304 “In conclusion” please delete it.

Please briefly write the key contents of the research results in the conclusion section.

Author Response

Reviewer 2

Introduction

Line 61 It feels sudden and needs to be modified more contextually and smoothly.

-> Line 63 (rephrase) For understanding the presence status of O. cf. ovata in Jeju coastal waters, we conducted to investigate spatial and seasonal distributions of the O. cf. ovata in Jeju, Korea.

Line 65-66 This MS does not mention environmental factors other than water temperature and salt, so please delete the contents of nutrients and DO.

->We corrected it in the text file as suggested.

Materials and Methods

Line 83 As in the introduction, delete pH and DO not mentioned in MS.

->We corrected it in the text file as suggested.

Line 92-94 Delete Method for Nutrient.

->We corrected it in the text file as suggested.

Line 99-101 Need to be clearly rewritten.

->We corrected it in the text file as suggested.

Line 100 . The fixed 300 ml of sample was placed still overnight to allow the cells to sink, and the supernatant was removed to adjust the final volume to 50-60 ml. In addition, the frozen macroalgae were weighed fresh after remove excess seawater

Line 112 Delete unnecessary characters in PDF file.

-> We corrected it in the text file as suggested.

Line 123-125 The target organism is an epiphytic organism, but did you just use seawater instead of shaking the seaweed for analysis? If so, there is a problem principally in the method.

-> We corrected it in the text file as suggested.

Line 122 To obtain DNA from field samples, 20–50 mL of samples obtained by shaking the macroalgae which collected from the sampling sites in the Jeju coastal water were filtered with Whatman GF/C (pore size 1.2 μm),

Line 160 Delete unnecessary characters in PDF file.

->We corrected it in the text file as suggested.

Results

Line 199 Delete unnecessary characters in PDF file.

->We corrected it in the text file as suggested.

Line 244 and Line 257 Figures 7 and 8 appear to contain the same information. Either one needs to be deleted, and it is recommended to delete Figure 7.

->We corrected it in the text file as suggested.

We deleted Figure 7

Discussion

Line 280-282 It is a sudden expression that is not natural in context. Insufficient to account for seasonal changes in target organisms through this reference.

->We corrected it in the text file as suggested.

Line 273 Delete sentence

Line 297 In Figure 9, it does not appear to be inhibited by lower salinity. The expression needs to be somewhat relaxed.

-> We corrected it in the text file as suggested.

Line 289 but also possible to play an inhibitory role by lowering salinity

Conclusion

Line 304 “In conclusion” please delete it.

Please briefly write the key contents of the research results in the conclusion section.

Incorporate at the end of the discussion
